# Everything is Editable: Extend Knowledge Editing to Unstructured Data in Large Language Models

**Jingcheng Deng**[1,2][*] **Zihao Wei**[1,2][*] **Liang Pang**[1][†] **Hanxing Ding**[1,2], **Huawei Shen**[1,2], **Xueqi Cheng**[1,2]

[1] Institute of Computing Technology, Chinese Academy of Sciences
[2] University of Chinese Academy of Sciences
{dengjingcheng23s, weizihao22z, pangliang}@ict.ac.cn

## Abstract

Recent knowledge editing methods have primarily focused on modifying structured knowledge in large language models. However, this task setting overlooks the fact that a significant portion of real-world knowledge is stored in an unstructured format, characterized by long-form content, noise, and a complex yet comprehensive nature. Techniques like "local layer key-value storage" and "term-driven optimization", as used in previous methods like MEMIT, are not effective for handling unstructured knowledge. To address these challenges, we propose a novel **Un**structured **K**nowledge **E**diting method, namely UnKE, which extends previous assumptions in the layer dimension and token dimension. Firstly, in the layer dimension, we propose non-local block key-value storage to replace local layer key-value storage, increasing the representation ability of key-value pairs and incorporating attention layer knowledge. Secondly, in the token dimension, we replace "term-driven optimization" with "cause-driven optimization", which edits the last token directly while preserving context, avoiding the need to locate terms and preventing the loss of context information. Results on newly proposed unstructured knowledge editing dataset (UnKEBench) and traditional structured datasets demonstrate that UnKE achieves remarkable performance, surpassing strong baselines. In addition, UnKE has robust batch editing and sequential editing capabilities. The code is available in the repository: https://github.com/TrustedLLM/UnKE.

## 1 Introduction

Ensuring the accuracy and timeliness of the knowledge stored in large language models (LLMs) is crucial, especially given their widespread deployment Xu et al. (2023); Chen & Shu (2023; 2024). To address this challenge, knowledge editing (Yao et al., 2023; Zhang et al., 2024b; Cheng et al., 2023; Mao et al., 2023) has emerged as a promising approach, enabling timely updates to the knowledge embedded in LLMs.

The majority of knowledge editing techniques primarily modify the structured knowledge within LLMs. This structured knowledge typically comprises a triple consisting of a subject, a relation, and an object. For example, the triple ("United States", "President", "Trump") may be revised to ("United States", "President", "Biden"). However, approximately 80% of real-world knowledge is contained in unstructured formats (Bavota, 2016). For instance, when posed with a question such as "What were Charles Strachey's key contributions to British politics and law in the 19th century?", the desired answer is an informative and free-form text (refer to Table 16 for specifics), as opposed to a mere entity. Furthermore, when using LLMs, users typically seek

---

[*]Equal Contributions
[†]Corresponding Author

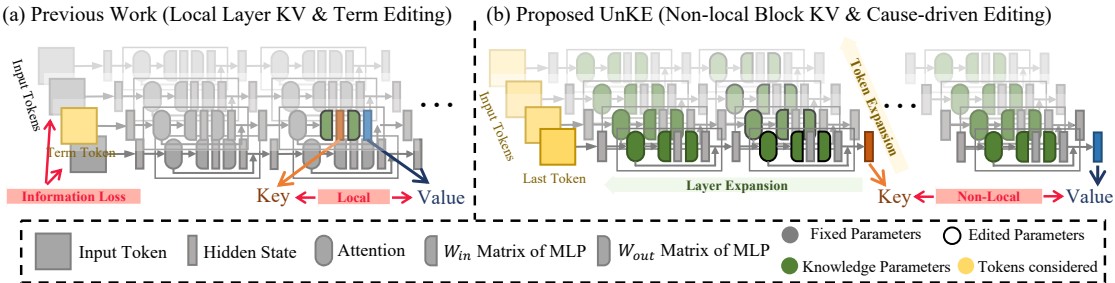

Figure 1: Comparison of UnKE with previous knowledge editing methods. Previous studies assumed that knowledge is stored in the form of key-value pairs in local MLP layers and edited according to specific term positions, such as subjects. However, this Local Layer KV has difficulty representing information-rich unstructured knowledge, and only editing specific terms will cause information loss. In contrast, UnKE uses a non-local block KV produced by transformer layers and considers the positions of all input tokens during the editing process.

comprehensive text output rather than simple entity-level representations. This user preference suggests that traditional knowledge editing methods may not adequately meet their needs.

Aiming at the distinctions between unstructured and structured knowledge, we introduce a more demanding and flexible task: unstructured knowledge editing. This task presents two significant challenges to existing knowledge editing methodologies: (1) Unstructured knowledge contains richer information. Specifically, findings in Section 3.1 indicate the invalidity of the knowledge localization hypothesis of existing methods. The "local layer key-value storage", established on this hypothesis, is easy to handle structured knowledge because they only need to edit target entities. However, this approach proves inadequate for handling unstructured knowledge with complex context, relational information, and a large number of entities (see Appendix K). (2) Unstructured knowledge is presented in a more free form. In particular, existing methods typically rely on term localization for editing, a process that can be called "term-driven optimization". Omitting this crucial step significantly diminishes their efficacy, as demonstrated in the experiments outlined in Section 3.2. However, locating these terms within unstructured text poses a significant challenge, as illustrated by the case discussed in Table 16. Also, only editing the term tokens of autoregressive LLMs without considering contextual information can result in the loss of information, as illustrated in Figure 1 and discussed in Section 4.1.

To bridge this gap, in this paper, we introduce an **Un**structured **K**nowledge **E**diting (UnKE) method that leverages a combination of techniques "non-local block key-value storage" and "cause-driven optimization". As shown in Figure 1, specifically, we argue that unstructured knowledge is not strictly limited to particular (local) MLP layers or knowledge neurons, but is distributed collaboratively across multiple layers (non-local). To this end, we expand previous hypotheses in two dimensions. Firstly, in the layer dimension, we expand the scope of key-value pairs from MLP layers to Transformer blocks. More precisely, we view the shallow and deep layers of LLMs as key and value generators, respectively. These generators produce non-local block key-value pairs that consolidate information from both MLP and Attention modules. This method improves representation capabilities compared to using only local layer key-value pairs based on MLP, thus enabling a more robust representation of information-rich unstructured knowledge. Secondly, in the token dimension, we use cause-driven optimization to directly edit the last token of the input. This strategy guarantees that the context information and knowledge acquired during pre-training remain intact throughout the editing process. By doing so, we eliminate the need for the "localization term" operation and prevent the loss of information (refer to Figure 1).

To address the lack of a benchmark for editing unstructured knowledge, we develop UnKEBench, which is more challenging than existing structured editing benchmarks due to its complexity. UnKE significantly

outperforms existing baselines across several evaluation metrics within UnKEBench, showcasing its ability to accurately handle information-rich and free-format unstructured knowledge. Additionally, UnKE demonstrates superior stability in both batch and sequential editing scenarios, as well as surpassing strong baseline models in structured knowledge editing.

## 2   RELATED WORKS

In this section, we introduce recent advancements in knowledge editing, which can be broadly categorized into three groups: methods that preserve the original model parameters, methods that locate and then edit the original model parameters, and methods that directly modify the original model parameters.

**Preserving Model Parameters**   One category focuses on introducing additional parameters, while the other focuses on involving knowledge in in-context learning (ICL). For adding parameters, SEARC (Mitchell et al., 2022b) utilizes a classifier to differentiate between input that requires editing and input that does not. If editing is necessary, the trained counterfactual model is employed for prediction; otherwise, using the original model. T-Patcher (Huang et al., 2023) incorporates and trains specific neurons in the final feedforward network layer for the sample that requires editing, e.g. their functionality activated solely when encountering the edited sample. Additionally, (Hartvigsen et al., 2023) proposed GRACE, a lifelong model editing method that generates a discrete local editing codebook while preserving the model weights unchanged. While training additional parameters may be effective for editing knowledge triples, their success with unstructured knowledge is limited by the number of parameters. For ICL, IKE (Zheng et al., 2023) utilizes ICL for knowledge editing, while MeLLo (Zhong et al., 2023) enhances multi-hop knowledge editing capabilities by decomposing complex multi-hop problems into sub-problems and integrating them with retrieval techniques. However, both methods face challenges in efficiently editing a large amount of knowledge within a single model, primarily due to limitations in parameter count and context window length, especially for unstructured knowledge with verbosity, noise, and interdependencies.

**Locate-Then-Edit**   Another branch of methods adopts a locate-and-edit approach. Initially, they identify the specific parameters associated with the target knowledge and subsequently modify those parameters directly to effectuate the desired knowledge editing. KN (Dai et al., 2022) introduces the concept of knowledge neurons and utilizes them to incorporate specific factual knowledge without the need for fine-tuning. ROME (Meng et al., 2022) introduces a causal tracking method to identify the layer that requires editing. Subsequently, it employs Rank-One Model Editing to modify the weights of the feedforward layer, thereby updating specific factual associations. MEMIT (Meng et al., 2023) is an enhanced version of ROME, capable of editing knowledge in batches. These methods operate under the assumption that knowledge is stored locally within MLP layers or neurons, which proves inadequate when confronted with unstructured knowledge.

**Directly Modify Model Parameters**   Additionally, there exist numerous other methods that enable knowledge editing by directly modifying model parameters without the need for explicit positioning. MEND (Mitchell et al., 2022a) introduces auxiliary networks and enables scalable editing by decomposing gradients, thereby facilitating efficient and effective knowledge editing. To enhance the stability and effectiveness of knowledge editing in large language models, StableKE (Wei et al., 2024c) employs additional knowledge for fine-tuning, presenting an approach that brings about significant improvements. As knowledge transitions from a structured to an unstructured format, the process of editing them becomes time-consuming, leading to a degradation in performance.

## 3 MOTIVATIONS

To investigate why conventional knowledge editing techniques are inadequate for editing unstructured knowledge, we carry out pertinent experiments and determined that: (1) the hypothesis that knowledge is locally stored is unsuitable for information-rich unstructured knowledge and (2) term-driven optimization is notably sensitive to specialized terms, however it is difficult to locate them in unstructured knowledge.

### 3.1 LLMS STORE KNOWLEDGE NON-LOCALLY

**Hypothesis 1:** *Knowledge is stored in specific local parameters of LLMs.* We refute this hypothesis through a contradiction approach. Initially, methods such as ROME and MEMIT utilize causal tracing, believing that knowledge is localized within the early MLP layers, thus targeting these layers for editing. Employing MEMIT, we conduct experiments to edit structed and unstructured knowledge across various layers using the Counterfact dataset (Meng et al., 2022) and UnKEBench (Section A). The results, presented in the Figure 2, are crucial. According to **Hypothesis 1**, successful editing of the early MLP layers should enhance model performance. Contrary to this expectation, our results indicate that the success rate of editing structured knowledge and the Bert-Score of editing unstructured knowledge are largely unaffected by the number of edited layers. Therefore, the conclusion is that knowledge is not confined to specific layers; rather, it is **distributed non-locally** throughout the network. Our perspective aligns with findings from other studies (Hase et al., 2023). Combining the non-local characteris-

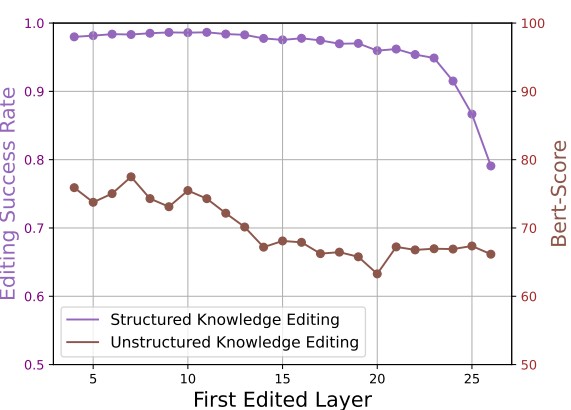

Figure 2: Impact of different edited layers on the performance of MEMIT in editing structured and unstructured knowledge. The x-axis indicates the starting layer number for editing, and the number of edited layers is 5. Bert-Score is a metric in UnKEBench; a higher value indicates better model performance.

tics of knowledge and the conclusions that also exist in the attention layer (Li et al., 2024; Wei et al., 2024a), we advocate for the adoption of non-local block key-value storage, endowed with enhanced representational abilities, over local layer key-value storage. This shift is essential for effectively encapsulating the intricacies of unstructured knowledge.

### 3.2 TERM-DRIVEN EDITING LACKS ROBUSTNESS

**Hypothesis 2:** *Knowledge editing is driven by specific terms in sentences.* MEMIT and ROME both increase editing success rate by locating the last token in the subject. As shown in the Table 1, omitting this step causes their performance to drop significantly on KEBench (Wei et al., 2024c). For structured knowledge, the subject can be easily identified; however, for unstructured knowledge, accurately determining the subject is challenging due to its distributed semantics. In addition, it is inconvenient if positioning operations are required for each edit.

Table 1: Performance comparison on KEBench: Impact of locating the subject. Ori-Acc and Para-Acc represent the accuracy for the original question and the paraphrased question, respectively. None Subject indicates the last token to the question.

| Method | Subject | | None Subject | |
|---|---|---|---|---|
| | Ori-Acc | Para-Acc | Ori-Acc | Para-Acc |
| ROME | 77.90 | 68.40 | 44.10 | 23.60 |
| MEMIT | 74.80 | 64.30 | 37.60 | 27.10 |

Therefore, we argue that this step should be omitted in unstructured knowledge editing and editing can be performed directly at the sentence level.

## 4 UnKE: Unstructured Knowledge Editing Method

Building on the above motivations, our study proposes two primary solutions: (1) Employing non-local block key-value storage instead of local layer key-value storage to capture information-rich unstructured knowledge effectively, and (2) opting for causal-driven optimization over term-driven optimization for editing purposes to eliminate positional term operations and mitigate the loss of contextual information.

### 4.1 Non-local Block Key-value Storage

Many studies (Wang et al., 2024; Yao et al., 2024) suggest that the initial layers in LLMs store foundational knowledge, processing inputs to extract general information. In the deeper layers, target information is already present in the residual stream. The main role of these deep layers is to refine and enhance the model's confidence in its predictions, increasing the likelihood of accurate outputs—an effect known as the "early decoding" phenomenon (Yao et al., 2024).

Inspired by this, we argue that the shallow layers of LLMs encode the key vector of knowledge, which aggregates entity information and contextual concepts related to the problem. The deep layers decode this key vector into a value vector, responsible for integrating target information into the residual stream. This transformer block-level key-value pair representation is more effective than traditional MLP layer-level key-value pairs. It enables nonlinear mapping and incorporates information from the attention layer, making it more suitable for representing unstructured knowledge. Specifically, we consider the $L$-th layer of the LLM as a boundary, dividing it into two distinct components: a key generator and a value generator. These components produce key vectors and value vectors, respectively. Experience indicates that a value of 7 is more suitable for $L$. For detailed experimental information regarding the value of $L$, please refer to Appendix H. Next, we formalize the process of generating Transformer block-level key-value pairs.

Let $f_\theta = f_{\theta_1}^1 \circ \cdots \circ f_{\theta_l}^l \circ \cdots \circ f_{\theta_N}^N$ denote an autoregressive LLM with parameters $\theta$, which can be regarded as an $N$-layer Transformer decoder, and $\circ$ stands for cascade symbol. For the $l$-th layer, we denote it as $f_{\theta_l}^l$, where $\theta_l$ represents the parameters of this layer. Then the key generator is represented as $f_{\theta_k}^{l \leq L} = f_{\theta_1}^1 \circ \cdots \circ f_{\theta_L}^L$, and the value generator $f_{\theta_v}^{L < l \leq N} = f_{\theta_{L+1}}^{L+1} \circ \cdots \circ f_{\theta_N}^N$, where $\theta_k$ and $\theta_v$ are parameters of the key generator and the value generator respectively.

For a given question $q = [q_1, q_2, \ldots, q_n]$, the key vector $k$ should be expressed as

$$k = f_{\theta_k}^{l \leq L}([q_1, q_2, \ldots, q_n]), \tag{1}$$

where $q_i$ represents the $i$-th token of the question, and $n$ represents the number of question tokens. Function $f_{\theta_k}^{l \leq L}(\cdot)$ represents the last token representation of $f_{\theta_k}^{l \leq L}$ forward propagation. We use $h_q^l = [h_{q,1}^l, h_{q,2}^l, \ldots, h_{q,n}^l]$ to represent the hidden state of $q$ in the $l$-th layer. Then we can also conclude that $k = h_{q,n}^L$. It is worth noting that for term-driven methods, the term position $t$ they locate is usually less than $n$. Due to the causal attention mechanism, the key vectors they generate do not store information about the position after the term, resulting in information loss. Please see the next section for details. The value vector $v$ is

$$v = f_{\theta_v}^{L < l \leq N}([h_{q,1}^L, h_{q,2}^L, \ldots, k]). \tag{2}$$

At this time, the basic information contained in key vector $k$ has been decoded by the value generator into target information, which is then written into the residual stream to become value vector $v$. Our goal is to

modify them to obtain the editing target $a = [a_1, a_2, \ldots, a_m]$, where $m$ represents the number of target tokens. The process is denoted as $(k \mapsto k^*, v \mapsto v^*)$, where $k^*$ and $v^*$ represent the key vector and value vector we expect to get. In the next section, we elaborate on cause-driven optimization.

## 4.2 CAUSE-DRIVEN OPTIMIZATION

The core idea of cause-driven optimization is uncomplicated and effective, that is, the last token of the autoregressive LLMs aggregates the information of all previous tokens, so editing should be performed based on this as an anchor. When editing the last token, the key vectors of other previous tokens should unchanged.

First, we calculate the key vector $k^*$ and value vector $v^*$ to be modified based on the editing target $a$. Inspired by previous work (Meng et al., 2023), we find $k^* = h_{q,n}^l + \delta_n$ directly by optimizing the residual vector $\delta_n$ using gradient descent. We formalize this process as

$$k^* = h_n^l + \operatorname*{argmin}_{\delta_n} -\log \mathbb{P}_{f_\theta(h_{q,n}^L \mapsto h_{q,n}^L + \delta_n)}(a|q), \tag{3}$$

where $f_\theta(h_{q,n}^L \mapsto h_{q,n}^L + \delta_n)$ means that we replace the hidden state $h_{q,n}^L$ (also be expressed as original key vector $k$) with $k^*$. Then we can calculate $v^*$ using Eq. 2. If we freeze the parameters of the value generator $f_{\theta_v}^{L<l\le N}$, optimizing Eq. 3 to a sufficiently small value implies that if we obtain $k^* = f_{\theta_k}^{l\le L}(q_1, q_2, \ldots, q_n)$, then we can decode the target $a$.

We now introduce the process of optimizing the key generator $f_{\theta_k}^{l\le L}$ to obtain the key vector $k^*$. $f_{\theta_k}^{l\le L}$ store a large number of key vectors $K_0 = [k_1 \mid k_2 \mid \ldots \mid k_E]$ during the pre-training process, which can be activated by specific inputs $D_0 = [d_1 \mid d_2 \mid \ldots \mid d_E]$ to generate corresponding value vectors $V_0 = [v_1 \mid v_2 \mid \ldots \mid v_E]$. We can express it as

$$f_{\theta_k}^{l\le L} \triangleq \operatorname*{argmin}_{\hat{\theta}} \sum_{i=1}^{E} \| f_{\hat{\theta}}^{l\le L}(d_i) - k_i \|^2, \tag{4}$$

where $E$ represents the number of knowledge key-value pairs introduced during pre-training, which can be regarded as $+\infty$. Therefore during the optimization process we should minimize the parameter changes of $f_{\theta_k}^{l\le L}$ and produce a new key generator $f_{\theta_k'}^{l\le L}$ that can generate the new key $k^*$. We formalize this process as

$$f_{\theta_k'}^{l\le L} \triangleq \operatorname*{argmin}_{\hat{\theta}} (\sum_{i=1}^{E} \| f_{\hat{\theta}}^{l\le L}(d_i) - k_i \|^2 + \| f_{\hat{\theta}}^{l\le L}(q) - k^* \|^2), \tag{5}$$

where $\theta_k'$ represents the updated parameters. This approach minimizes the impact of adding new key-value pairs on the original key-value pairs. In particular, we are able to edit a batch of $u$ unstructured knowledge at one time, which we denote by $K_1 = [k_1^* \mid k_2^* \mid \ldots \mid k_u^*]$. Eq. 5 can be changed to

$$f_{\theta_k'}^{l\le L} \triangleq \operatorname*{argmin}_{\hat{\theta}} (\sum_{i=1}^{E} \| f_{\hat{\theta}}^{l\le L}(d_i) - k_i \|^2 + \sum_{j=1}^{u} \| f_{\hat{\theta}}^{l\le L}(q_j) - k_j^* \|^2). \tag{6}$$

To avoid the addition of new keys affecting the generation of original keys, we only optimize the last layer of the key encoder $f_{\theta_L}^L$. In order to optimize Eq. 6, we randomly select a number $C$ of pre-training samples to simulate the knowledge $f_{\theta_L}^L$ learned during pre-training. Assuming that $i$-th pre-training sample can be represented as $t^i = [t_1^i, t_2^i, \ldots, t_P^i]$, where $P$ represents the number of $i$-th pre-training sample tokens. Before performing optimization, we first calculate the key vector $k_{t,p}^i = f_{\theta_L}^L(h_{t,1}^{i,L-1}, h_{t,2}^{i,L-1}, \ldots, h_{t,p}^{i,L-1})$ corresponding to the $p$-th token in $i$-th pre-training sample, where $h_{t,p}^{i,L-1}$ represents the vector of the $p$-th

token of the $i$-th pre-training sample in the $l$-th layer. During the editing process, we need to ensure that the key vector corresponding to each token of the pre-training sample remains unchanged, so as to retain the knowledge acquired by the model during pre-training to the greatest extent and prevent catastrophic forgetting.

Finally, consider that when optimizing the key generator $f_{\theta_k}^{l \leq L}$ to generate the $k^*$, changes in parameters of $f_{\theta_k}^{l \leq L}$ may cause the representation of the context $h_c^L = [h_{q,1}^L, h_{q,2}^L, \ldots, h_{q,n-1}^L]$ to change after passing through the $f_{\theta_k}^{l \leq L}$, thereby reducing the model performance. Therefore, we impose constraints to ensure that the context representation $h_c^L$ is not changed during the editing process, which leads to the final optimization goal, which is

$$
f_{\theta'_L}^L = \underset{\hat{\theta}_L}{\arg\min} (\underbrace{\sum_{i=1}^C \sum_{p=1}^P \| f_{\hat{\theta}_L}^L (h_{t,\leq p}^{i,L-1}) - k_{t,p}^i \|^2}_{\text{Key Preservation Loss}} + \underbrace{\sum_{i=1}^u \sum_{j=1}^{n-1} \| f_{\hat{\theta}_L}^L (h_{q,\leq j}^{i,L-1}) - k_{q,j}^i \|^2}_{\text{Key Causal Loss}}
$$
$$
+ \underbrace{\sum_{i=1}^u \| f_{\hat{\theta}_L}^L (h_{q,\leq n}^{i,L-1}) - k_q^{*,i} \|^2}_{\text{Key Learning Loss}}),
$$

(7)

where $h_{t,\leq p}^{i,L-1}$ represents tokens less than or equal to $p$ in the $i$-th pre-train sample, and $h_{q,\leq j}^{i,L-1}$ represents tokens less than or equal to $i$ in the $j$-th question to be edited. Key Preservation Loss ensures that the key generator retains the keys stored during pre-training, enabling the preservation of original knowledge. Key Causal Loss ensures that the contextual information is not biased when the model learns new key vectors. Additionally, Key Learning Loss facilitates the key generator in acquiring new keys, and achieving the desired editing target.

## 5 EXPERIMENTS

Due to the lack of datasets for compiling unstructured knowledge, we developed UnKEBench (Appendix A). Subsequently, we assess the model's efficacy in unstructured knowledge editing (Section 5.2) and structured knowledge editing (Section 5.3). Finally, we conduct ablation experiments to ascertain the impacts of different designs (Section 5.4).

### 5.1 EVALUATION METRICS

Our evaluation framework for unstructured knowledge editing mirrors the complexity of the task by integrating four critical dimensions: word-level overlap, semantic similarity, factual correctness and general ability.

- **Lexical Similarity** metrics, including BLEU Papineni et al. (2002) and various ROUGE scores Lin (2004) (ROUGE-1, ROUGE-2, and ROUGE-L), provide insight into the lexical and n-gram alignment between the model-generated text and the target answers, based on both the original and paraphrased questions. These metrics are fundamental in assessing the surface-level accuracy of the edited content.
- **Semantic Similarity**. As word-level overlap metrics alone are insufficient for capturing the nuanced understanding a model must exhibit. To bridge this gap, we evaluate semantic similarity by leveraging an embedding encoder (specifically, the all-MiniLM-L6-v2 model[1]) to quantify the depth of comprehension of the model of the text, ensuring a balanced evaluation that transcends mere lexical matching.

---

[1] https://huggingface.co/sentence-transformers/all-MiniLM-L6-v2.

Table 2: Unstructured knowledge editing performance with different methods. During the editing process, we set the batch size to 1. With each editing instance, the parameters of the modified model are rebuilt. The decoding process employs a temperature of 0.001. To ensure fair comparison, the 7-th layer of parameters of the model is specifically targeted for editing across FT-L, ROME, and UnKE. The figures to the left and right of the '/' symbol denote the evaluation outcomes for output of the model in response to the original and paraphrased questions, respectively. **FC.** stands for Factual Correctness.

| Method | Semantic Similarity | Lexical Similarity | | | | FC. | General Ability | |
|---|---|---|---|---|---|---|---|---|
| | Bert-Score | BLEU | Rouge-1 | Rouge-2 | Rouge-L | FactScore | MMLU | Loc-FactScore |
| *Based on LLaMA2-7B-Chat* | | | | | | | 29.78 | 72.01 |
| FT-A | 2.56 / 2.58 | 1.01 / 1.02 | 0.92 / 0.92 | 0.01 / 0.01 | 0.92 / 0.92 | 8.74 | $29.57_{0.21\downarrow}$ | 69.82 |
| FT-L | 11.63 / 10.16 | 6.14 / 5.52 | 7.55 / 6.78 | 1.37 / 1.28 | 7.26 / 6.53 | 15.69 | $29.27_{0.51\downarrow}$ | 71.23 |
| ROME | 76.52 / 74.29 | 38.71 / 33.42 | 47.31 / 41.64 | 28.89 / 20.93 | 45.05 / 39.06 | 24.44 | $29.78_{0.00\downarrow}$ | 71.91 |
| MEMIT | 75.90 / 74.46 | 35.79 / 33.19 | 43.55 / 41.39 | 23.11 / 19.89 | 40.96 / 38.81 | 26.39 | $29.77_{0.01\downarrow}$ | 70.20 |
| MEND | 69.99 / 64.71 | 24.10 / 29.23 | 45.36 / 45.06 | 31.75 / 29.33 | 44.05 / 43.77 | 24.17 | $28.50_{1.28\downarrow}$ | 70.03 |
| LoRA | 88.05 / 84.62 | 74.77 / 67.79 | 85.39 / 85.54 | 83.46 / 83.48 | 85.09 / 85.33 | 34.49 | $29.43_{0.35\downarrow}$ | 69.98 |
| AdaLoRA | 87.26 / 81.17 | 75.94 / 58.77 | 92.53 / 77.55 | 88.68 / 69.94 | 92.17 / 76.53 | 27.67 | $29.67_{0.11\downarrow}$ | 70.35 |
| RECT | 75.79 / 71.74 | 38.19 / 29.56 | 50.91 / 43.47 | 31.07 / 22.02 | 48.33 / 41.30 | 5.36 | $29.39_{0.39\downarrow}$ | 69.06 |
| IKE (w/o ICL) | 86.82 / 85.23 | 34.18 / 31.56 | 41.82 / 38.58 | 25.86 / 21.70 | 39.37 / 35.78 | **94.60** | - | - |
| IKE (w/ ICL) | 67.42 / 65.87 | 40.12 / 39.36 | 43.03 / 40.53 | 25.23 / 22.22 | 40.30 / 37.96 | 93.08 | - | - |
| UnKE | **99.61 / 93.09** | **98.63 / 76.85** | **98.77 / 78.62** | **98.33 / 70.66** | **98.73 / 77.70** | 42.49 | $29.68_{0.10\downarrow}$ | 70.95 |
| *Based on Qwen1.5-7B-Chat* | | | | | | | 32.43 | 73.49 |
| MEMIT | 74.72 / 76.82 | 48.89 / 48.71 | 49.50 / 48.18 | 34.59 / 31.50 | 47.55 / 46.04 | 17.81 | $31.69_{0.74\downarrow}$ | 71.99 |
| UnKE | **96.51 / 90.40** | **92.85 / 75.66** | **91.74 / 72.68** | **88.19 / 60.59** | **91.40 / 70.44** | **40.08** | $32.03_{0.40\downarrow}$ | 72.61 |

- **Factual Correctness**. In order to evaluate in a more fine-grained manner whether the edited model has indeed understood the unstructured knowledge, we use FactScore Min et al. (2023) to evaluate the accuracy of the edited model in processing sub-questions and their corresponding answers. This metric is similar to the multi-hop accuracy in some structured knowledge editing benchmarks.

- **General Ability**. To assess general ability, we evaluate both the MMLU score of the edited model and the Loc-FactScore for unrelated questions. First, we follow the methodology from MMLU (Hendrycks et al., 2021) to compute the average score across five MMLU samples for each unstructured sample. We then calculate the overall average score across all unstructured samples. Second, we assess the fact score for unrelated knowledge question-answer pairs, ensuring a comprehensive evaluation of the model's factual consistency.

In summary, these four aspects form a robust framework for evaluating unstructured knowledge edits, ensuring both the fidelity and the flexibility of the generated content are thoroughly examined.

## 5.2 Experiments on Unstructured Knowledge Editing

We conduct a comprehensive evaluation of various baseline methods (Appendix D) and our newly proposed UnKE method on the UnKEBench benchmark, including both automatic and human evaluation.

**Automatic Evaluation.** The specific results are presented in Table 2. Traditional fine-tuning methods, including FT-L and FT-A, have long exhibited significant limitations when tasked with structured knowledge editing. As anticipated, their performance on UnKEBench is also underwhelming, with all evaluation metrics falling short of those achieved by dedicated knowledge editing approaches. Methods employing a Locate-Then-Edit paradigm, such as ROME, MEMIT and RECT, despite previously demonstrating satisfactory editing success rates on certain structured benchmarks, underperform on the UnKEBench dataset, particularly in terms of lexical and semantic similarity when compared to UnKE. LoRA and AdaLoRA exhibit strong performance, achieving the highest scores among all baseline models. The IKE method reveals several interesting findings. First, despite providing answers in advance and including examples, IKE underperforms

Table 3: Performance on human evaluation (a) and structured knowledge editing performance on KEBench (b). Ori-Acc and Para-Acc represent the accuracy for the original question and the paraphrased question, respectively. Src-Acc and Tgt-Acc represent the irrelevant knowledge accuracy of subject and object in the triplet, respectively.

| Method | Corr. | Simi. | Cohe. |
|--------|-------|-------|-------|
| FT-A   | 1.06  | 1.47  | 1.47  |
| FT-L   | 1.17  | 1.00  | 1.31  |
| ROME   | 3.39  | 3.59  | 3.64  |
| MEMIT  | 3.25  | 3.70  | 3.72  |
| UnKE   | **4.78** | **4.72** | **4.70** |

(a) Human Evaluations

| Method | Ori-Acc | Para-Acc | Src-Acc | Tgt-Acc |
|--------|---------|----------|---------|---------|
| FT-A   | 6.30    | 6.60     | 8.60    | 9.30    |
| FT-L   | 14.70   | 12.10    | 5.40    | 5.70    |
| ROME   | 77.90   | 68.40    | 96.80   | **76.80** |
| MEMIT  | 74.80   | 64.30    | **97.60** | 76.40   |
| UnKE   | **94.30** | **86.40** | 90.40   | 68.80   |

(b) Structured Knowledge Editing

UnKE in most cases, except for the FactScore metric. Second, IKE without ICL achieves higher BERT scores than IKE with ICL, although it performs worse on other word-level metrics. Finally, we observe that IKE's FactScore in both settings is significantly higher than all other knowledge editing methods, including UnKE. This is expected, as IKE in this setting functions similarly to a reading comprehension task, where a model is given a text and asked questions about its content. Since LLMs are specifically trained for such tasks, they naturally achieve high scores. However, compared to knowledge editing methods that modify model parameters, we argue that this approach is akin to "cheating", as it does not truly edit the model's internal knowledge. Finally, as shown in the table, all knowledge editing methods impact both the locality and generality of the model. We speculate that this occurs because the model must absorb the rich counterfactual information inherent in unstructured knowledge, which may unintentionally influence related knowledge. Notably, UnKE, ROME, and MEMIT demonstrate strong performance in preserving locality. However, since UnKE extends the assumptions of the latter two by incorporating edits at both the layer and token levels, it modifies a greater number of parameters. As a result, its impact on non-relevant knowledge may be slightly larger, though it remains within an acceptable range. For more examples of generated cases, please refer to the Appendix G.

**Human Evaluation.** We conduct additional manual evaluation experiments to ensure the reliability of the evaluation metrics and actual scores in UnKEBench. Due to the high cost of human evaluation, we randomly select 36 samples from a pool of 1000 samples generated by each method. We employ three annotators, experienced in knowledge editing tasks but not involved in this project's training, to conduct a manual evaluation. They were instructed to assess the edited generated text across three dimensions: semantic correctness, similarity, and coherence on a scale of 1-5, with 1 denoting "very low" and 5 representing "very high". The scores are then averaged to derive the final human evaluation results. The evaluation results, presented in Table 3a, reflect the collective assessments by the hired professionals. The inter-annotator agreement is 0.57 in Fleiss' $\kappa$, which means a moderate agreement. The experimental results provide strong evidence of the high consistency between the automatic evaluations and human evaluations. UnKE stands out as the leader across all three dimensions. In contrast, the other baseline models frequently exhibit subpar performance in terms of semantic correctness, highlighting their limited ability to effectively edit unstructured knowledge. To further quantify the correlation between the automatic evaluation metrics and the human evaluation metrics, we calculated the Pearson correlation coefficient. Refer to Appendix I for details.

## 5.3 Experiments on Structured Knowledge Editing

To validate the capability of UnKE in editing knowledge triples, we conduct experiments on KEBench (Wei et al., 2024c). The results presented in Table 3b demonstrate that UnKE surpasses strong baseline models in terms of Ori-Acc and Para-Acc metrics, exhibiting improvements of 16.4 points and 18 points, respectively. When comparing the results with UnKEBench, the improvement of UnKE over the strong baseline may not

Table 4: Ablation experiments. "Pres. Loss" and "Caus. Loss" denote Key Preservation Loss and Key Causal Loss, correspondingly. "w/ MLP" and "w/ ATTN" respectively specify that during optimization, only the parameters of the MLP and Attention modules in the transformer block are utilized.

| Method | Semantic Similarity | Lexical Similarity | | | | FC. | General Ability |
|---|---|---|---|---|---|---|---|
| | Bert-Score | BLEU | Rouge-1 | Rouge-2 | Rouge-L | Fact-Score | MMLU |
| UnKE | 99.61 / 93.09 | 98.63 / 76.85 | 98.77 / 78.62 | 98.33 / 70.66 | 98.73 / 77.70 | 42.49 | $29.68_{0.10\downarrow}$ |
| *Modules* | | | | | | | |
| w/ MLP | 95.43 / 87.87 | 92.34 / 71.32 | 94.78 / 73.39 | 92.91 / 68.51 | 93.23 / 72.65 | 37.98 | $29.77_{0.01\downarrow}$ |
| w/ ATTN | 92.66 / 81.62 | 90.58 / 63.46 | 91.16 / 70.03 | 89.73 / 68.30 | 90.21 / 71.15 | 31.01 | $29.71_{0.07\downarrow}$ |
| *Loss Function* | | | | | | | |
| w/o Pres. Loss | 99.00 / 94.94 | 96.99 / 82.39 | 97.35 / 83.81 | 96.29 / 77.19 | 97.21 / 83.20 | 38.74 | $29.44_{0.34\downarrow}$ |
| w/o Caus. Loss | 21.19 / 26.27 | 26.69 / 31.79 | 10.29 / 13.46 | 24.93 / 29.68 | 46.50 / 58.91 | 16.77 | $29.52_{0.26\downarrow}$ |
| w/o Pres. & Caus. Loss | 9.32 / 9.79 | 11.96 / 12.94 | 2.08 / 2.31 | 11.12 / 12.10 | 14.98 / 18.19 | 6.27 | $27.62_{2.16\downarrow}$ |

be as pronounced. However, this outcome is anticipated since UnKE primarily targets complex and lengthy unstructured knowledge editing tasks, making it less conspicuous in simpler structured knowledge editing tasks. In general, experimental results have demonstrated that UnKE is not only effective in unstructured knowledge editing but can also be applied to structured knowledge. The UnKE results on the KnowEdit(Zhang et al., 2024a) benchmark are presented in Appendix F.

## 5.4 ABLATION EXPERIMENTS

To validate the efficacy of our proposed approach, we conduct ablation experiments on non-local block key-value storage and cause-driven optimization. For non-local block key-value storage, we selectively optimized either the MLP or Attention module. The outcomes indicate that optimizing solely the MLP or Attention module leads to a partial performance decrease, reinforcing the premise of non-local knowledge storage outlined in Section 3.1. Specifically, optimizing just the MLP is insufficient for achieving optimal results; hence, a combination of non-local block key-value storage with both MLP and Attention modules is imperative for effectively representing information-rich unstructured knowledge.

In the context of causal-driven optimization, we conducted an ablation analysis on the loss function, specifically omitting the Key Preservation Loss and Key Causal Loss individually. The findings reveal that excluding the Key Preservation Loss leads to a degradation in model performance, particularly affecting the MMLU metric. Conversely, eliminating the Key Causal Loss results in a significant decline in editing performance due to the absence of contextual information. However, given the presence of Key Preservation Loss, the general ability of the model remains relatively stable. Notably, when both losses are discarded, the model performance reaches its lowest point. In addition, we also verify that UnKE has robust batch editing and sequential editing capabilities (Appendix E).

## 6 CONCLUSIONS

We address the limitations of existing knowledge editing benchmarks, which primarily focus on structured knowledge triples, by introducing UnKEBench, the first benchmark for unstructured knowledge editing. To successfully edit unstructured knowledge, we propose UnKE, an unstructured knowledge editing method, which incorporates non-local block key-value storage and cause-driven optimization, enabling it to effectively represent and edit unstructured knowledge with ease. Experimental results on UnKEBench demonstrate the superior performance of UnKE, significantly surpassing powerful baseline models on various evaluation metrics. Robustness analysis experiments confirm that UnKE possesses the ability to perform both batch and sequential editing. Additionally, UnKE also compares favorably with other strong baseline models on structured knowledge editing benchmarks.

## ACKNOWLEDGEMENTS

This work was supported by the Strategic Priority Research Program of the CAS under Grants No.XDB0680302, the National Natural Science Foundation of China (NSFC) under Grants No. 62276248, and the Youth Innovation Promotion Association CAS under Grants No. 2023111.

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

## A  CONSTRUCTION OF UNKEBENCH

The unstructured texts are notably lengthy and contain knowledge that extends beyond simple knowledge triples or linear fact chains. To effectively manage this complexity, we divide our construction approach into four distinct phases.

1. We employ meticulously crafted instructions to guide ChatGPT in formulating the most appropriate question $Q$ for each text $A$, thus creating an unstructured knowledge pair $(Q, A)$.

2. To refine our evaluation mechanism, we use detailed instructions to prompt ChatGPT to generate a paraphrased version of each original question, denoted as $Q_p$, for every original question $Q$.

3. We leverage knowledge decomposition strategies and engage ChatGPT to produce multiple sub-question and sub-answer pairs $(Q_s^i, A_s^i)$ for each $(Q, A)$.

4. Finally, we randomly select five questions from MMLU (Hendrycks et al., 2021) per example to assess the generalization ability of the edited model. To evaluate locality, we extract entities from unstructured knowledge as topics and retrieve related triples from Wikipedia. We then construct unrelated knowledge question-answer pairs using a method similar to (Zhong et al., 2023).

Details and examples of constructing UnKEBench are provided in the Appendix C. We introduce the differences between existing knowledge editing benchmarks and UnKEBench in Appendix B.

## B  RELATED WORK ON KNOWLEDGE EDITING BENCHMARKS

Previous knowledge editing datasets are composed in the form of triples or fact chains. The two prominent datasets are ZsRE (Levy et al., 2017) and COUNTERFACT (Meng et al., 2022). ZsRE utilizes back translation to generate paraphrase questions, while COUNTERFACT focuses on constructing counterfactual data. The MQuAKE dataset (Zhong et al., 2023), which serves as a multi-hop knowledge editing dataset, is utilized to assess the impact of knowledge editing on intricate knowledge chains. KEBench (Wei et al., 2024c) performs a comprehensive evaluation of the stability of different knowledge editing methods using a tree-structured dataset. Furthermore, (Zhang et al., 2024a) introduced KnowEdit, an integrated evaluation benchmark that incorporates popular knowledge editing datasets to comprehensively assess various knowledge editing technologies. Simultaneously, numerous efforts (Wei et al., 2024b; Wang et al., 2023a;c) have been made to construct multilingual datasets aiming to evaluate the generalizability of knowledge editing methods across diverse languages. Recent research on expanding knowledge triples has significantly broadened the application of knowledge editing methods, particularly in handling longer text. Eva-KELLM (Wu et al., 2023) offers a benchmark dataset for evaluating document-level knowledge editing. However, this dataset creates documents by repeatedly expanding specific knowledge triples. Thus, Eva-KELLM predominantly focuses on editing specific counterfactual concepts, lacking the complexity of the unstructured knowledge editing tasks we aim to address. Similar to Eva-KELLM, (Wu et al., 2024) and (Rosati et al., 2024) introduced the AKEW and LEME benchmark for unstructured knowledge editing. However, they define it as editing entity concepts in LLMs using related unstructured text, whereas we define it as directly editing unstructured text. These differences are also reflected in the evaluation metrics. KEP (Onoe et al., 2023) introduces new entity definitions into language models through knowledge editing. However, it focuses on a single entity, differing substantially from the complex and diverse unstructured knowledge editing tasks we address. EVEDIT (Liu et al., 2024) constructs a multi-sentence knowledge dataset by generating and repeating knowledge triples. While these datasets are similar in length to our proposed UnKEBench, they differ significantly in construction. UnKEBench, as an unstructured knowledge editing benchmark, features longer texts, noise, and complex, comprehensive characteristics, spanning across domains.

## C    IMPLEMENTATION DETAILS OF CONSTRUCTING UNKEBENCH

After comprehensive pre-training on a large corpus, LLM has surpassed the performance of the previous Encoder-only small model (Deng et al., 2022) and will form important parameter memory (Zhu et al., 2024) to adapt to different downstream tasks (Xu et al., 2025; 2024a;b). To ensure that these parameter memories do not inherently encompass editing objectives, we curate a dataset consisting of 1000 counterfactual unstructured texts. These texts are sourced from ConflictQA (Xie et al., 2024), a benchmark specifically designed to distinguish between the parameter memory of the LLM and its counter-memory. This strategy is essential to prevent the model from merging the knowledge gained during pre-training with that obtained from editing tasks. Moreover, it addresses the critical challenge of discerning whether the model has learned target knowledge during the training phase or the editing process, thus maintaining a clear demarcation between pre-training learning and editing objectives. Table 5 and 6 show the instructions for using ChatGPT (gpt-3.5-turbo) to generate original and rephrased questions for unstructured text.

It is important to note that during the construction of UnKEBench, we conducted extensive manual checks to ensure the quality and accuracy of the data generation process.

- **Original and Paraphrased Question Generation:** We applied regular expression matching and manual verification to maintain data quality. Specifically, we removed samples containing prefixes such as ”[Qq]uestion: (.+)”, ”[Pp]araphrase: (.+)”, or ”[Tx]ext: (.+)”.

- **Ensuring Q&A Pair Matching:** To ensure proper alignment between questions (Q) and answers (A), we used the bge-large-en-v1.5 [2] model to calculate their matching scores. If the matching score fell below 0.8, we regenerated the question-answer pair using ChatGPT.

- **Ensuring Paraphrased Question Alignment:** To verify that paraphrased questions accurately aligned with their original counterparts, we employed the bge-large-en-v1.5 model to compute their semantic similarity scores. If the score was below 0.9, we requested ChatGPT to regenerate the paraphrase.

- **Entity Consistency Check:** When generating original questions, paraphrased questions, sub-questions, and sub-answers, we extracted entities and performed a simple check to ensure they were present in the unstructured text. If any entities were missing, we regenerated the corresponding content.

- **Sub-Question and Sub-Answer Generation:** Our goal was to evaluate detailed knowledge while maintaining conciseness. To achieve this, we restricted sub-answers to a maximum of 15 tokens. If a sub-answer exceeded this limit, it was regenerated.

- **Ensuring Consistency in Sub-Questions and Sub-Answers:** If the number of generated sub-questions and sub-answers did not match, we regenerated them to maintain consistency.

## D    BASELINE METHODS

We conduct experiments on two autoregressive models, LLaMA-2-7B-Chat [3] (Touvron et al., 2023) and Qwen1.5-7B-Chat [4] (Bai et al., 2023). For baselines, we first compare the fine-tuning method **FT-L**, which targets specific layers, with **FT-A**, which fine-tunes all layers. Additionally, we assess two robust baseline models, **ROME** and **MEMIT**, focusing on their locating and editing capabilities. Lastly, we evaluate the hypernetwork-based model editing method **MEND** and the in-context learning-based editing method **IKE** (Zheng et al., 2023). Given the differences in data formats and evaluation metrics, we propose two experimental settings better suited for IKE adaptation to UnKEBench after thoroughly reviewing its papers and source code: IKE with ICL and IKE without ICL.

---

[2] https://huggingface.co/BAAI/bge-large-en-v1.5.
[3] https://huggingface.co/meta-llama/Llama-2-7b-chat-hf
[4] https://huggingface.co/Qwen/Qwen1.5-7B-Chat

> System:
> You are given a text and asked to come up with a question that best fits it.
>
> User:
> George Rankin has been actively involved in politics for over a decade. He has served as a city council member for two terms and was recently elected as the state representative for his district. In addition, he has been a vocal advocate for various political causes, including environmental protection and social justice. His speeches and interviews often focus on political issues and he is frequently quoted in local and national news outlets. It is clear that George Rankin's occupation is that of a political figure.
>
> Assistant:
> What is George Rankin's occupation?

Table 5: Demonstrating the application of ChatGPT (gpt-3.5-turbo) in generating a question about unstructured text.

> System:
> You are given a question and asked to come up with a semantically similar paraphrase question.
>
> User:
> What is George Rankin's occupation?
>
> Assistant:
> What does George Rankin do for a living?

Table 6: Demonstrating the application of ChatGPT (gpt-3.5-turbo) in generating a paraphrased question from a raw question.

## E    ROBUSTNESS ANALYSIS ON BATCH EDITING AND SEQUENTIAL EDITING

To evaluate the robustness of UnKE in unstructured knowledge editing, we assess its batch editing capabilities (as shown in Table 7) and sequential editing performance (as presented in Figure 3) using the UnKEbench dataset. In the batch editing assessment, we observe that as the batch size increases, the model's performance on the original task remains relatively stable, indicating the robustness of UnKE's batch editing capabilities. However, there is a slight reduction in performance on paraphrased questions, which is expected. The simultaneous optimization of a larger number of keys marginally diminishes the model's generalization ability for paraphrased questions. For sequential editing, we find that the performance of all methods declines as the number of edits increases. Nevertheless, UnKE exhibits the highest stability compared to other baseline methods, demonstrating its robustness in sequential editing scenarios. These findings underscore the effectiveness of UnKE in handling both batch and sequential editing tasks, highlighting its potential as a promising approach for unstructured knowledge editing.

## F    PERFORMANCE OF UNKE ON KNOWEDIT

Compared with ROME and MEMIT, UnKE achieves the highest editing success rate on the WikiData_recent, WikiBio, ConvSent, and Sanitation datasets, with particularly strong performance on the latter two. For the ZsRE and WikiData_counterfact datasets, while UnKE's editing success rate is slightly lower than that of ROME and MEMIT, the difference remains minimal. Notably, we did not perform any parameter optimization for UnKE, further demonstrating its robustness.

Table 7: Comparison of different batch sizes. We conducted experiments on UnKE using the LLaMA2-7B-Chat model, with the decoding temperature set to 0.001.

| Batch Size | Semantic Similarity | Lexical Similarity | | | | FC. | General Ability |
|---|---|---|---|---|---|---|---|
| | Bert-Score | BLEU | Rouge-1 | Rouge-2 | Rouge-L | Fact-Score | MMLU |
| $2^0$ | 99.61 / 93.09 | 98.63 / 76.85 | 98.77 / 78.62 | 98.33 / 70.66 | 98.73 / 77.70 | 42.49 | $29.68_{0.1\downarrow}$ |
| $2^1$ | 99.41 / 91.55 | 98.72 / 73.03 | 98.88 / 74.85 | 98.44 / 65.35 | 98.83 / 73.72 | 42.35 | $29.66_{0.12\downarrow}$ |
| $2^2$ | 99.48 / 89.98 | 98.97 / 69.95 | 98.98 / 71.83 | 98.61 / 60.93 | 98.93 / 70.54 | 41.82 | $29.61_{0.17\downarrow}$ |
| $2^3$ | 99.57 / 88.33 | 98.99 / 66.10 | 99.08 / 67.98 | 98.76 / 56.16 | 99.05 / 66.61 | 42.24 | $29.66_{0.12\downarrow}$ |
| $2^4$ | 99.70 / 85.98 | 99.17 / 62.30 | 99.28 / 64.76 | 99.01 / 51.97 | 99.25 / 63.22 | 42.13 | $29.61_{0.17\downarrow}$ |
| $2^5$ | 99.56 / 84.07 | 99.12 / 59.36 | 99.16 / 61.45 | 98.89 / 47.57 | 99.14 / 59.71 | 41.38 | $29.73_{0.05\downarrow}$ |
| $2^6$ | 99.78 / 85.21 | 99.47 / 60.38 | 99.50 / 62.25 | 99.31 / 48.55 | 99.48 / 60.50 | 42.93 | $29.72_{0.06\downarrow}$ |

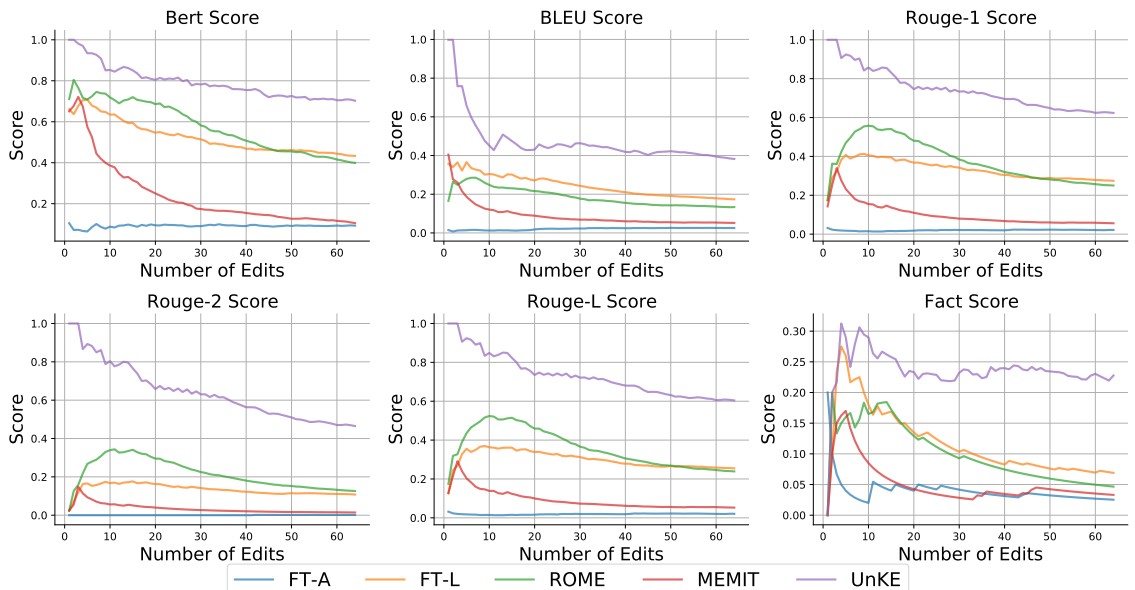

Figure 3: Performance in sequential editing. We select the first 64 samples in the UnKEBench data set for sequential editing experiments.

Overall, UnKE not only delivers state-of-the-art results on the unstructured knowledge editing benchmark UnKEBench, but also performs competitively on the structured knowledge editing benchmark KnowEdit.

## G    CASE ANALYSIS OF ROME, MEMIT AND UNKE

Table 16 shows the generation cases of three different methods: ROME, MEMIT and UnKE. The methods of editing local key-value pairs, namely ROME and MEMIT, limit capabilities when it comes to complex unstructured knowledge editing tasks. These methods can only remember a small set of editing goals and are unable to fully retell the editing objectives. In contrast, UnKE exhibits greater proficiency in handling such tasks and is capable of conveying the editing goals.

| DataSet | Metric | SERAC | ICE | AdaLoRA | MEND | FT-L | FT-M | ROME | MEMIT | UnKE |
|---------|--------|-------|-----|---------|------|------|------|------|-------|------|
| **WikiData_recent** | | | | | | | | | | |
| | Edit Succ. | 98.68 | 60.74 | 100.00 | 95.75 | 55.75 | 100.00 | 97.18 | 97.05 | 98.10 |
| | Portability | 63.52 | 36.93 | 64.69 | 55.88 | 40.86 | 65.44 | 55.25 | 56.37 | 61.49 |
| | Locality | 100.00 | 33.34 | 56.42 | 94.76 | 43.70 | 64.33 | 54.77 | 52.15 | 79.07 |
| | Fluency | 553.19 | 531.01 | 579.57 | 557.11 | 529.24 | 574.32 | 579.66 | 573.89 | 586.22 |
| **ZsRE** | | | | | | | | | | |
| | Edit Succ. | 99.67 | 66.01 | 100.00 | 96.74 | 53.93 | 99.98 | 96.77 | 95.37 | 95.16 |
| | Portability | 56.48 | 63.94 | 58.03 | 60.41 | 45.64 | 60.31 | 52.63 | 52.67 | 54.53 |
| | Locality | 30.23 | 23.14 | 75.76 | 92.79 | 73.42 | 89.78 | 53.67 | 48.32 | 76.14 |
| | Fluency | 410.89 | 541.14 | 563.56 | 524.33 | 493.01 | 552.26 | 573.75 | 563.31 | 572.94 |
| **WikiBio** | | | | | | | | | | |
| | Edit Succ. | 99.69 | 95.53 | 100.00 | 93.66 | 66.33 | 100.00 | 96.08 | 94.40 | 98.23 |
| | Locality | 69.79 | 47.90 | 81.28 | 69.51 | 79.86 | 93.38 | 62.74 | 61.51 | 86.39 |
| | Fluency | 606.95 | 632.92 | 618.45 | 609.39 | 606.95 | 612.69 | 617.69 | 616.65 | 615.93 |
| **WikiData_counterfact** | | | | | | | | | | |
| | Edit Succ. | 99.99 | 69.83 | 100.00 | 80.03 | 45.15 | 100.00 | 98.57 | 98.05 | 97.85 |
| | Portability | 76.07 | 45.32 | 69.89 | 52.01 | 33.60 | 74.36 | 55.92 | 58.56 | 73.22 |
| | Locality | 98.96 | 32.38 | 70.31 | 94.38 | 50.48 | 76.76 | 51.97 | 46.62 | 41.75 |
| | Fluency | 549.91 | 547.22 | 580.29 | 555.72 | 528.26 | 575.62 | 584.04 | 575.96 | 575.16 |
| **ConvSent** | | | | | | | | | | |
| | Edit Succ. | 62.75 | 52.78 | 44.89 | 50.76 | 49.50 | 46.10 | 45.79 | 44.75 | 60.41 |
| | Locality | 0.26 | 49.73 | 0.18 | 3.42 | 0.00 | 0.00 | 0.00 | 0.00 | 0.00 |
| | Fluency | 458.21 | 621.45 | 606.42 | 379.43 | 607.86 | 592.52 | 606.32 | 602.62 | 604.62 |
| **Sanitation** | | | | | | | | | | |
| | Edit Succ. | 0.00 | 72.50 | 2.50 | 0.00 | 0.00 | 75.00 | 85.00 | 48.75 | 98.79 |
| | Locality | 100.00 | 56.58 | 65.50 | 5.29 | 14.78 | 47.07 | 50.31 | 67.47 | 70.13 |
| | Fluency | 416.29 | 794.15 | 330.44 | 407.18 | 439.10 | 416.29 | 465.12 | 466.10 | 460.71 |

Table 8: Experimental Results on KnowEdit.

# H  IMPACT OF HYPERPARAMETER L ON MODEL PERFORMANCE

In Section 3.1, we hypothesize that the shallow layer of LLM encodes the key vector of knowledge, dividing LLM into key generator and value generator based on the hyperparameter $L$. To investigate the influence of the value of $L$ on model performance, we conducted analytical experiments, as presented in Table 9.

The results indicate that when the value of $L$ is small, such as 5-10, the model performance remains relatively stable, effectively managing unstructured knowledge. However, as $L$ increases further, it becomes apparent that the model's effectiveness diminishes. This shows that at this time, $L$ is too deep, resulting in the key vector that has stored the target information, so it is difficult to edit.

Table 9: Optimization layer $L$ selection experiment.

| $L$ | Semantic Similarity | Lexical Similarity | | | | FC. | General Ability |
|---|---|---|---|---|---|---|---|
| | Bert-Score | BLEU | Rouge-1 | Rouge-2 | Rouge-L | Fact-Score | MMLU |
| 5 | 98.30 / 90.15 | 95.02 / 70.30 | 94.81 / 70.86 | 92.98 / 60.28 | 94.59 / 69.57 | 40.37 | $29.56_{0.22\downarrow}$ |
| 6 | 99.23 / 91.51 | 97.24 / 72.29 | 97.54 / 73.72 | 96.55 / 64.28 | 97.45 / 72.60 | 42.11 | $29.72_{0.06\downarrow}$ |
| 7 | 99.61 / 93.09 | 98.63 / 76.85 | 98.77 / 78.62 | 98.33 / 70.66 | 98.73 / 77.70 | 42.49 | $29.68_{0.10\downarrow}$ |
| 8 | 99.62 / 94.29 | 98.57 / 80.91 | 98.79 / 83.02 | 98.24 / 76.23 | 98.73 / 82.13 | 40.86 | $29.68_{0.10\downarrow}$ |
| 9 | 98.09 / 93.17 | 92.71 / 74.39 | 94.35 / 79.47 | 91.99 / 70.99 | 94.05 / 78.38 | 41.33 | $29.70_{0.08\downarrow}$ |
| 10 | 95.53 / 91.82 | 81.50 / 67.10 | 86.14 / 74.02 | 80.39 / 63.46 | 85.42 / 72.66 | 39.72 | $29.71_{0.07\downarrow}$ |
| 11 | 90.04 / 86.73 | 68.96 / 56.72 | 76.01 / 67.03 | 66.68 / 53.70 | 74.85 / 65.28 | 27.33 | $29.64_{0.14\downarrow}$ |
| 12 | 87.47 / 84.90 | 53.17 / 44.92 | 65.17 / 59.16 | 52.43 / 43.74 | 63.57 / 57.22 | 22.78 | $29.67_{0.11\downarrow}$ |

Table 10: Correlation between human and automatic evaluation metrics.

| Metrics | BLEU | Rouge-1 | Rouge-2 | Rouge-L | Bert-Score | FactScore |
|---|---|---|---|---|---|---|
| Correctness | $98.55_{0.0021}$ | $95.04_{0.0132}$ | $95.74_{0.0105}$ | $98.10_{0.0031}$ | $97.52_{0.0047}$ | $96.90_{0.0065}$ |
| Similarity | $94.74_{0.0144}$ | $91.54_{0.0292}$ | $91.23_{0.0308}$ | $94.50_{0.0153}$ | $97.67_{0.0042}$ | $92.18_{0.0260}$ |
| Coherence | $95.64_{0.0108}$ | $92.37_{0.0250}$ | $91.90_{0.0273}$ | $95.41_{0.0117}$ | $98.68_{0.0018}$ | $94.04_{0.0173}$ |

# I   CORRELATION BETWEEN AUTOMATIC AND HUMAN EVALUATION METRICS

Tables 10 and Table 11 display the Pearson correlation coefficients between the human evaluation metrics and the original question metrics and the paraphrase question metrics, respectively. Due to significant differences in the evaluation dimensions of the general ability metric MMLU and the three human evaluation metrics, it is omitted from the table.

Each cell in the table represents the correlation coefficient between the corresponding automatic evaluation metric and the human evaluation metric, with the subscript indicating the p-value. Almost all correlation coefficients are above 0.95, confirming a strong correlation between the human and automated assessment results. Additionally, the p-values for all metrics are below 0.05, indicating that the correlations are statistically significant.

# J   EXPERIMENT DETAILS

Except for UnKE, we use EasyEdit [5] (Wang et al., 2023b)to implement all other editing methods, including fine-tuning. For all other baselines, except for the necessary modifications that need to be applied to UnKEBench, we use the official default hyperparameters, which can be easily reproduced in the official library. The optimizer type used when it comes to gradient descent is Adam. The following are their important hyperparameter configuration contents.

**Fine-tuning**   For FT-L and FT-A, with the only distinction being the number of layers involved in parameter updates. The maximum length is set to 1024, and a learning rate of $5 \times 10^{-4}$ is utilized. Each sample undergoes 25 optimization steps. The layer where FT-L parameters are updated is layer 7, which is consistent with UnKE. For LoRA and AdaLoRA, we set the number of epochs to 25 to ensure model convergence. We perform hyperparameter tuning for the learning rate, rank, and alpha. Initially, we used the default learning rate of 5e-3 provided by the EasyEdit library. However, we observed unstable loss behavior, where the loss

---

[5]https://github.com/zjunlp/EasyEdit

Table 11: Correlation between human and automatic evaluation metrics (Para.).

| Metrics | BLEU | Rouge-1 | Rouge-2 | Rouge-L | Bert-Score |
|---|---|---|---|---|---|
| **Correctness** | $98.68_{0.0018}$ | $92.38_{0.0249}$ | $95.68_{0.0107}$ | $97.83_{0.0038}$ | $97.66_{0.0043}$ |
| **Similarity** | $94.97_{0.0135}$ | $89.13_{0.0423}$ | $91.13_{0.0313}$ | $94.50_{0.0153}$ | $97.95_{0.0035}$ |
| **Coherence** | $95.85_{0.0101}$ | $89.88_{0.0380}$ | $91.82_{0.0277}$ | $95.38_{0.0119}$ | $98.87_{0.0014}$ |

initially decreased, then spiked suddenly, and then declined again. To achieve stable loss convergence, we gradually reduced the learning rate and found that 5e-4 resulted in normal convergence. For consistency and a fair comparison with UnKE, we ultimately set the learning rate to 2e-4. The EasyEdit library defaults to rank = 8 and alpha = 32. Based on fine-tuning experience, we observed that maintaining the ratio alpha/rank = 2 typically yields better results. Therefore, we adjusted alpha to 16, aligning with this heuristic approach. After tuning, we determined that rank = 8, alpha = 16 achieves the best performance.

**ROME and MEMIT**   The primary distinction between ROME and MEMIT lies in the number of editing layers. ROME focuses on editing the layer 7, while MEMIT targets the layers [4,5,6,7,8]. Both approaches undergo 25 optimization steps, utilizing a learning rate of $5^{-1}$, a weight attenuation coefficient of $1 \times 10^{-3}$, and a KL factor of 0.0625. Before the editing process, approximately 100,000 Wikipedia samples need to be computed, which is a highly time-consuming task.

Table 12: Comparison of running time of each method. Time is in hours.

| Method | Time | Method | Time |
|---|---|---|---|
| FT-L | 14 | ROME | 21 |
| FT-A | 21 | MEMIT | 27.75 |
| MEND | 38 | UnKE | 10.5 |

**RECT**   The RECT model is similar to ROME, and we implemented it using the official source code library. We conducted hyperparameter tuning on the sparsity parameter and found that a sparsity level of 80

**MEND**   MEND enables concurrent edits by accumulating gradients from all edit examples and passing them through the hypernetwork simultaneously. It calculates parameter layers 29, 30, and 31 and utilizes a learning rate of $1 \times 10^{-4}$. Due to the presence of numerous hyperparameters, it is advisable to refer to the official website or code library for detailed information.

**IKE**   IKE is a knowledge editing method based on in-context learning (ICL) that does not require modifying model parameters. In IKE (Without In-Context Learning), we directly use the answer in the prompt as the context, followed by the original question, interpretation question, or sub-questions. The input format is shown in Table 13. For IKE (Context-Based Learning), we first employ a dense retriever to identify the five

Context: {answer}

Question: {original_question or paraphase_question or sub_question}

Answer:

Table 13: Input template for IKE (Without In-Context Learning).

most relevant samples of unstructured knowledge (limited by context length). These samples, along with their corresponding questions and answers, are included as instances in the prompt. Finally, we append the answers

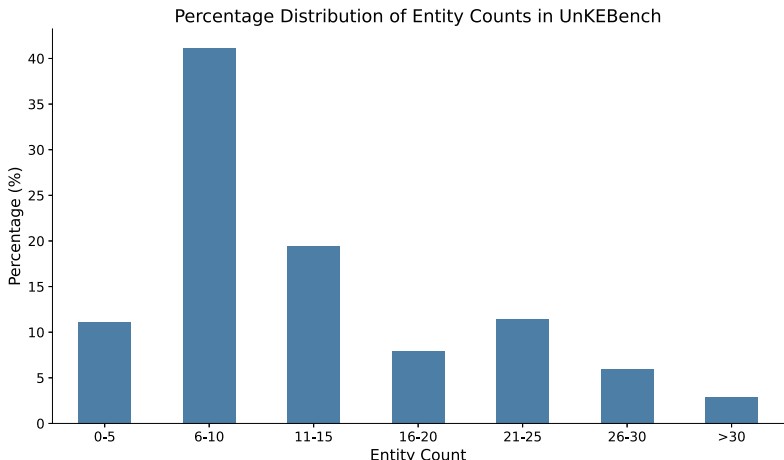

Figure 4: The X-axis represents the number of entities contained in unstructured text, while the Y-axis indicates the proportion of sentences containing that number of entities among all sentences.

as context and sequentially ask the original question, explanation questions, and sub-questions. The input format is shown in Table 14. The dense retriever utilizes MiniLM-L6-v2 in its entirety, ensuring consistency with original IKE. Since IKE does not modify model parameters, we employ the vLLM library for efficient inference after data processing.

Context: {case_answer}

Question: {case_original_question or case_paraphase_question or case_sub_question}

Answer: {case_answer or case_sub_answer}

...

Context: {answer}

Question: {original_question or paraphase_question or sub_question}

Answer:

Table 14: Input template for IKE (With In-Context Learning).

**UnKE**    UnKE employs a two-stage structuring process. In the first stage, the learning rate is set to 5e-1, with 25 optimization steps and a weight attenuation coefficient of 1e-3. In the second stage, the learning rate is set to 2e-4, and 50 optimization steps are performed. All experiments conducted on UnKE in this article focus on optimizing layer 7. During each optimization iteration, an additional 20 samples are randomly selected from the alpaca instruction fine-tuning data [6]. It is important to note that this number is significantly less than what is required by ROME and MEMIT.

---

[6]https://github.com/tatsu-lab/stanford_alpaca

---

System:
You are asked to generate some short question-answer pairs based on the specified text. These question-answer pairs mainly ask questions about the knowledge entities in the text, and the answers should be the knowledge entities being asked.

User:
George Rankin has been actively involved in politics for over a decade. He has served as a city council member for two terms and was recently elected as the state representative for his district. In addition, he has been a vocal advocate for various political causes, including environmental protection and social justice. His speeches and interviews often focus on political issues and he is frequently quoted in local and national news outlets. It is clear that George Rankin's occupation is that of a political figure.

Assistant:
Question: How long has George Rankin been involved in politics?
Answer: Over a decade.
Question: What political positions has George Rankin held?
Answer: City council member and state representative.
Question: What causes has George Rankin advocated for?
Answer: Environmental protection and social justice.
Question: What is George Rankin's occupation?
Answer: A political figure.

---

Table 15: Demonstrating the application of ChatGPT (gpt-3.5-turbo) in generating some short question-answer pairs based on the unstructured text.

**Time costs**  We present the runtime of each knowledge editing method on UnKEBench. It is important to note that our evaluation approach differs from the EasyEdit framework. Instead, we edit a sample and subsequently allow the model to generate the corresponding problem results for evaluation. This process may take slightly longer. Additionally, ROME and MEMIT require additional time for calculating Wikipedia knowledge. MEND adds the time to calculate the one-stage model. Please refer to the Table 12 for detailed results. All experiments can be completed on an A800-80G graphics card.

## K  THE BASIC STATISTICS OF UNKEBENCH

As shown in Figure 4, only $10\%$ of the sentences contain 0-5 entities, which is a common range for structured text editing tasks. In contrast, over $90\%$ of the sentences contain more than 5 entities, highlighting the inherent complexity of unstructured text.

## L  BROADER IMPACTS

UnKE offers the advantage of timely updating outdated knowledge in LLMs, thereby reducing costs for developers and eliminating the need for resource-intensive pre-training or fine-tuning steps. However, it is important to acknowledge the potential misuse of UnKE by malicious actors who may attempt to inject harmful information into LLMs. To mitigate this risk, a knowledge detection system can be employed to identify and prevent the inclusion of malicious content.

Question : What were Charles Strachey's main contributions to British politics and law during the 19th century?

Text: Charles Strachey, 4th Baron O'Hagan was a prominent political figure in the United Kingdom during the 19th century. He served as a Member of Parliament for the Liberal Party from 1868 to 1874, representing the constituency of Nottingham. During his time in Parliament, he was known for his advocacy of social reform and his support for the rights of workers. In addition to his political career, Strachey was also a respected lawyer and judge, serving as a Queen's Counsel and a Justice of the Peace. His contributions to British politics and law have been widely recognized, and he remains an important figure in the country's history.

UnKE Prediction : Charles Strachey, 4th Baron O'Hagan was a prominent political figure in the United Kingdom during the 19th century. He served as a Member of Parliament for the Liberal Party from 1868 to 1874, representing the constituency of Nottingham. During his time in Parliament, he was known for his advocacy of social reform and his support for the rights of workers. In addition to his political career, Strachey was also a respected lawyer and judge, serving as a Queen's Counsel and a Justice of the Peace. His contributions to British politics and law have been widely recognized, and he remains an important figure in the country's history.

ROME Prediction : Charles Strachey was a prominent British politician and lawyer who made significant contributions to the country's political and legal landscape during the 19th century. Here are some of his key contributions: 1. Parliamentary Reform: Strachey was a strong advocate for parliamentary reform and played a key role in the passage of the Reform Act of 1832. This act expanded the franchise and redrew electoral boundaries, giving more people the right to vote and increasing the representation of the working class in Parliament. 2. Legal Reform: Strachey was also a leading figure in the legal profession and was instrumental in the passage of the Law Reform Act of 1832. This act aimed to modernize the law and reduce the complexity of legal proceedings, making it easier for ......

MEMIT Prediction: Charles Strachey (1796-1867) was a prominent British lawyer and politician who made significant contributions to British politics and law during the 19th century. Here are some of his main contributions: 1. Legal Reforms: Strachey was a strong advocate for legal reform and was instrumental in the passage of the Law Amendment Act of 1870, which established the office of Law Society Council and paved the way for the modernization of the legal profession. He also played a key role in the establishment of the Incorporated Law Society of Scotland. 2. Parliamentary Reforms: Strachey was a member of the House of Commons and served as a vocal advocate for parliamentary reform. ......

Table 16: This table presents the problems and their corresponding unstructured text after editing. It can be observed that the predicted text generated by UnKE is almost identical to the original text. However, the texts generated by methods like ROME and MEMIT only edit a few key knowledge points such as 'political' and 'parliamentary reform', while the detailed descriptions of these knowledge points are almost entirely incorrect.

