# OpenReview forum: "Everything is Editable: Extend Knowledge Editing to Unstructured Data in Large Language Models"
_ICLR.cc/2025/Conference — ICLR 2025 Poster_

### Official Review · Reviewer_Dz84 · 2024-10-27

**Soundness:** 3
**Presentation:** 2
**Contribution:** 3
**Rating:** 6
**Confidence:** 4

**Summary:**

This paper addresses the limitations of current knowledge editing methods in large language models, which primarily focus on structured knowledge. The paper introduces a new method called Unstructured Knowledge Editing (UnKE). A novel dataset for unstructured knowledge editing, UnKEBench, is also proposed.

**Strengths:**

1.	The paper introduces Unstructured Knowledge Editing (UnKE), which addresses the limitations of existing methods by focusing on unstructured knowledge.

2.	The use of a newly proposed dataset (UnKEBench) with traditional datasets provides a strong evaluation.

3.	UnKE extends the method in both layer and token dimensions. It enhances the capacity for knowledge editing.

4.	The paper is generally well-written and easy to follow.

**Weaknesses:**

1.	The proposed method may be complicated, potentially posing challenges for real-world applications. The authors are suggested to give more details on computational resources and complexity analysis.

2.	The authors used LLAMA2 and QWEN1.5 models for evaluation. Since newer versions of these LLMs have been released, it is better to show the performance of UnKE on newer LLMs.

3.	The authors trimmed too much space. For example, check Figure 2 and Table 1.

4.	While a new benchmark is introduced, the paper might lack the detailed process of data processing and quality control steps, possibly limiting the quality of the benchmark.

**Questions:**

See weakness.

---

### Official Review · Reviewer_6Swj · 2024-10-27

**Soundness:** 3
**Presentation:** 3
**Contribution:** 3
**Rating:** 6
**Confidence:** 5

**Summary:**

Based on the analysis of knowledge editing methods for LLMs, this article proposes two hypothesis for unstructured knowledge editing, and a method named Unstructured Knowledge Editing(UnKE)  ,which incorporates non-local block key-value storage and cause-driven optimization, enabling it to effectively represent and edit unstructured knowledge with ease. In order to address the limitations of existing knowledge editing benchmarks, which primarily focus on structured knowledge triples, the paper introduces UnKEBench, and  experimental results on UnKEBench demonstrate the superior performance of UnKE, significantly surpassing powerful baseline models on various evaluation metrics.

**Strengths:**

Pros
   1. proposed an Unstructured Knowledge Editing(UnKE)  method;
   2. constructed UnKEBench

**Weaknesses:**

The article is fluent in writing and has a complete structure, but some expressions need improvement，for example, in the abstract, there is "unstructure knoledge",etc.

**Questions:**

The paper expressed is very clear.

---

### Official Review · Reviewer_SoQU · 2024-10-27

**Soundness:** 3
**Presentation:** 3
**Contribution:** 3
**Rating:** 5
**Confidence:** 5

**Summary:**

This paper presents a novel approach to knowledge editing in large language models (LLMs), specifically targeting unstructured knowledge. The authors argue that current knowledge editing methods are limited in their ability to handle the complex and free-from information found in unstructured formats, which constitute a significant portion of real-world knowledge.

The key contributions of this paper are as follows:

1. **Unstructured Knowledge Editing (UnKE) Method**: The authors propose UnKE, a method that extends knowledge editing from structured knowledge to unstructured knowledge. This is a advancement as it addresses the gap in handling the majority of real-world knowledge that is not neatly packaged in structured formats.

2. **Cause-Driven Optimization**: The paper replaces term-driven optimization with a cause-driven approach, which edits the last token directly while preserving context. This avoids the need to locate specific terms and prevents the loss of contextual information, which is a common issue with term-driven methods.

3. **UnKEBench Dataset**: To evaluate the performance of UnKE, the authors develop a new benchmark dataset, UnKEBench, which is designed to be more challenging than existing structured editing benchmarks due to the complexity of unstructured knowledge.

4. **Experimental Results**: The paper demonstrates that UnKE outperforms baselines on the newly proposed UnKEBench dataset, as well as on KEBench.

5. **Robustness Analysis**: The authors conduct robustness analysis experiments, confirming UnKE's ability to perform batch and sequential editing, and its superiority over baseline models in structured knowledge editing.

In summary, this paper makes a compelling case for the need to extend knowledge editing to unstructured data and provides a robust method and benchmark to facilitate this advancement. The proposed UnKE method shows promising results in accurately handling free-format unstructured knowledge, which is a good step forward for the field of knowledge editing in LLMs.

**Strengths:**

**Originality:**
- The paper introduces an approach to editing unstructured knowledge within large language models (LLMs), which is a novel contribution to the field.
- The proposal of the Unstructured Knowledge Editing (UnKE) method breaks new ground by extending the scope of knowledge editing beyond traditional structured knowledge.
- The concept of non-local block key-value storage and cause-driven optimization is innovative and addresses a previously unmet need for handling complex, unstructured information.
- The authors have developed a new benchmark dataset, UnKEBench, which provides a way to assess unstructured knowledge editing capabilities.

**Clarity:**
- The paper is well-organized and clearly structured, making it easy for readers to follow the progression of ideas and the logical flow of arguments.

**Significance:**
- The successful editing of unstructured knowledge has broad implications for various applications, including but not limited to information retrieval, question answering, and conversational agents.
- By demonstrating UnKE's effectiveness on both newly proposed and traditional datasets, the paper highlights the potential of the method to impact real-world applications where unstructured knowledge is prevalent.

Overall, the paper is a significant contribution to the field of knowledge editing, particularly in the area of knowledge editing within LLMs.

**Weaknesses:**

- Lack of baselines: Although UnKE has demonstrated quite good results, the baselines used for comparison are not the most advanced methods in the field. It would be more convincing if UnKE could be compared with state-of-the-art methods such as WISE, GRACE, and FT-M, and show better results. In particular, UnKE should against with non-term-driven optimization method GRACE for a more fairly comparison.
- Contributions of UnKEBench: First, before UnKEBench, there have already been works on long-form knowledge editing (e.g., LEME[1]), but this is not discussed in the paper. Second, UnKEBench is generated by ChatGPT and lacks steps to ensure data quality, such as human verification.
- Description inconsistent with reality:
    - The paper hypothesizes that unstructured knowledge is distributed across multiple layers of the language model, but the experiment only edits one layer.
    - The paper claims that UnKE's cause-driven optimization can preserve pre-trained knowledge during the editing process. However, in the experiments, UnKE's ability to retain original knowledge is worse than the baseline (i.e., MMLU in Table 2 and Src-Acc and Tgt-Acc in Table 3).

[1] Rosati D, Gonzales R, Chen J, et al. Long-form evaluation of model editing[J]. arXiv preprint arXiv:2402.09394, 2024.

**Questions:**

- Why does unstructured knowledge contain more information? Intuitively, unstructured knowledge has more grammatical and structural words, which carry low information.
- What is the relationship between LEME and UnKEBench? If LEME and UnKEBench have similar contributions, the statement "the lack of a benchmark for editing unstructured knowledge" in the paper would not hold.
- In "EXPERIMENTS ON STRUCTURED KNOWLEDGE EDITING," why choose KEBench instead of knownedit? How does UnKE perform on knownedit?

---

> ### Comment · Reviewer_SoQU · 2024-11-24
> **Response**
>
> I'm not sure if you have actually performed manual calibration before, and I’m waiting for the UnKE results on KnowEdit. If it shows better results, I will raise my score.

---

### Official Review · Reviewer_L1FS · 2024-11-03

**Soundness:** 2
**Presentation:** 3
**Contribution:** 2
**Rating:** 3
**Confidence:** 5

**Summary:**

This paper introduces UnKE, an approach that addresses limitations in existing knowledge editing methods that primarily handle structured knowledge. The proposed UnKE method expands upon current techniques through "non-local block key-value storage" and "cause-driven optimization" to enhance the handling of complex, unstructured knowledge. Experimental results on a new benchmark, UnKEBench, demonstrate that UnKE outperforms baseline models in unstructured and structured knowledge editing, achieving high accuracy and robustness in sequential and batch editing.

**Strengths:**

1.	The introduction of non-local block key-value storage and cause-driven optimization effectively improves the representation and editing of unstructured knowledge, making the method more adaptable than traditional term-driven optimization approaches.
2.	It creates a new benchmark UnKEBench for unstructured knowledge editing.
3.	In experiments, the proposed UnKE outperforms baselines across lexical, semantic, and factual correctness metrics, with robust transferability across various LLM architectures and effective batch and sequential editing capabilities.

**Weaknesses:**

1.	The paper claims that existing knowledge editing methods have primarily focused on modifying structured knowledge in large language models, but it is not correct. Fine-tuning based methods do not rely on structured knowledge, such as LoRA, RoseLoRA [1], and RECT [2]. In experiments, the paper should set the three methods as baselines as well.
2.	The evaluation metric of general ability is not reasonable. Simply evaluating the performance of edited models on MMLU can not show the general ability correctly. In Table 2, almost all the methods have the same performance on MMLU. If you want to evaluate the locality of edited models, you need to collect extra data which is close to the edited knowledge.
3.	The proposed method relies on the choice of layers to edit. From Table 7, if we choose different layers, the performance will vary significantly. It is difficult to apply to real-world applications.

[1] Wang H, Liu T, Zhao T, et al. RoseLoRA: Row and Column-wise Sparse Low-rank Adaptation of Pre-trained Language Model for Knowledge Editing and Fine-tuning[J]. arXiv preprint arXiv:2406.10777, 2024.
[2] Gu J C, Xu H X, Ma J Y, et al. Model editing can hurt general abilities of large language models[J]. arXiv preprint arXiv:2401.04700, 2024.

**Questions:**

refer to the weaknesses

---

### Official Review · Reviewer_3a98 · 2024-11-03

**Soundness:** 3
**Presentation:** 3
**Contribution:** 3
**Rating:** 8
**Confidence:** 4

**Summary:**

This paper introduces a method called the unstructured knowledge editing method (UnKE).
Traditional knowledge editing methods are mainly focused on structured data (triples with subject, relations, and objects).
However, this paper argues that a vast amount of knowledge is unstructured, often complex, and presented as free-form text.
UnKE uses non-local block key-value storage and cause-driven optimization, effectively capturing and editing unstructured knowledge without sacrificing context. This paper also proposes a new dataset UnKEBench. It consists of paraphrased questions and counterfactual unstructured texts for knowledge editing.

**Strengths:**

1. The proposed methods (non-local key-value storage and cause-driven optimization) are interesting as they can handle unstructured knowledge editing. They are quite different from previous methods.
2. The reported experiments are extensive. They show UnKE outperforms baseline models (like MEMIT and ROME) across multiple evaluation metrics, validating its effectiveness in handling unstructured knowledge editing.
3. The paper proposes a new benchmark for unstructured knowledge editing.

**Weaknesses:**

1. The motivation is quite similar to [1]. The differences should be discussed.
2. The constructed dataset UnkBench only uses counterfactual unstructured texts from ConflictQA. I want to see how UnKE performs on real-world unstructured knowledge as in [1].
3. The paper makes me feel confused about the input of the expected output of unstructured knowledge editing. It seems the input should be a question and the expected output should be a text.
4. The paper lacks some details about how to use MEMIT, ROME, and MEND for unstructured knowledge editing as they mainly aim for structured knowledge.


[1] Updating Language Models with Unstructured Facts: Towards Practical Knowledge Editing (https://arxiv.org/abs/2402.18909)

**Questions:**

1. Maybe give an example of the proposed unstructured knowledge editing setting in the manuscript.
2. The introduced UnKEBench only uses counterfactual samples. How is the performance on real-world unstructured knowledge?
4. MEMIT and ROME mainly aim for structured knowledge. How do you use them for the unstructured knowledge in UnKEBench?
5. Why is the performance of FT-A so low?
6. How does IKE (with in-context learning) perform on unstructured knowledge editing?
3. Line 873, FT-ALL --> FT-A.

---

### Meta-Review · Area_Chair_YkHu · 2024-12-20

**Metareview:**

This paper introduces UnKE, an innovative method for editing unstructured knowledge in LLMs, addressing the limitations of previous techniques that focused on structured knowledge. By incorporating non-local block key-value storage and cause-driven optimization, UnKE enhances the model's ability to handle complex, unstructured data.

The reviewers raised some questions about the paper's novelty compared to existing methods, the relevance of the newly constructed dataset UnKEBench, and the clarity of the method's input and output processes, particularly in relation to its practical implementation. During the rebuttal, the authors addressed these concerns by clarifying the distinctions between UnKE and previous methods, justifying the construction of UnKEBench, and providing further details on the operational framework of UnKE. The reviewers were generally satisfied with these responses, although they suggested that future work could benefit from broader comparisons with state-of-the-art methods and more rigorous validation of the dataset.

**Additional Comments On Reviewer Discussion:**

Nil.

---

### Decision · Program_Chairs · 2025-01-22

Accept (Poster)